# Clinical Significance of the Histone Deacetylase 2 (HDAC-2) Expression in Human Breast Cancer

**DOI:** 10.3390/jpm12101672

**Published:** 2022-10-08

**Authors:** Nikolaos Garmpis, Christos Damaskos, Dimitrios Dimitroulis, Gregory Kouraklis, Anna Garmpi, Panagiotis Sarantis, Evangelos Koustas, Alexandros Patsouras, Iason Psilopatis, Efstathios A. Antoniou, Michail V. Karamouzis, Konstantinos Kontzoglou, Afroditi Nonni

**Affiliations:** 1Second Department of Propedeutic Surgery, Laiko General Hospital, Medical School, National and Kapodistrian University of Athens, 11527 Athens, Greece; 2N.S. Christeas Laboratory of Experimental Surgery and Surgical Research, Medical School, National and Kapodistrian University of Athens, 11527 Athens, Greece; 3Renal Transplantation Unit, Laiko General Hospital, 11527 Athens, Greece; 4Department of Surgery, Evgenideio Hospital, Medical School, National and Kapodistrian University of Athens, 11527 Athens, Greece; 5First Department of Propedeutic Internal Medicine, Laiko General Hospital, Medical School, National and Kapodistrian University of Athens, 11527 Athens, Greece; 6Molecular Oncology Unit, Department of Biological Chemistry, Medical School, National and Kapodistrian University of Athens, 11527 Athens, Greece; 7Second Department of Pulmonology, Sotiria General Hospital, 11527 Athens, Greece; 8Charité-Universitätsmedizin Berlin, Corporate Member of Freie Universität Berlin and Humboldt-Universität zu Berlin, Augustenburger Platz 1, 13353 Berlin, Germany; 9First Department of Pathology, Medical School, National and Kapodistrian University of Athens, 11527 Athens, Greece

**Keywords:** breast, cancer, clinicopathological, deacetylase, HDAC, histone

## Abstract

Background/Aim: There is a strong association between malignancy and histone deacetylases (HDACs). Histone deacetylase inhibitors (HDACIs) are now being tested as antitumor agents in various clinical trials. We aimed to assess the clinical importance of HDAC-2 in breast cancer (BC). Materials and Methods: A total of 118 BC specimens were examined immunohistochemically. A statistical analysis was conducted in order to examine the relation between HDAC-2 and the clinicopathological features and survival of the patients. Results: Higher HDAC-2 expression was related to lobular histological type of cancer, grade III, and stage III BC. In addition, the disease-free period and overall survival were curtailed and negatively related to the over-expression of HDAC-2. Other factors correlating with worse survival were histological types other than ductal or lobular, and the stage of the disease. Conclusions: This study showed a relationship between HDAC-2 and BC. Further studies are required in order to eventually potentiate the role of HDACIs as anticancer agents in BC.

## 1. Introduction

The regulation of transcription is controlled by the acetylation status of histones. The competition between the histone deacetylases (HDACs) and the histone acetyltransferases (HATs) plays a crucial role in either the inhibition or the promotion of transcription, respectively [1,2]. The HDAC-dependent removal of acetyl groups from lysine residues in histones leads to the formation of heterochromatin, thus repressing the process of transcription [3]. A large number of human HDACs has been found and recognized. HDACs are divided into four classes depending on their structure and function [4], and are further subdivided into NAD-dependent (III) and Zn2+-dependent classes (I, II, and IV). Class I includes HDAC-1, -2, -3, and -8; class II consists of HDAC-4, -5, -7, -6, -9, and -10, whereas class IV only contains HDAC-11. Class III HDACs are widely known as sirtuins and contain seven distinct members: SIRT-1–7 (Figure 1) [5,6,7].

In particular, HDAC-2 is a classical class I HDAC with a conserved deacetylase domain with short amino- and carboxy-terminal extensions [3]. Its catalytic site consists of a 14 Å long internal cavity adjacent to the zinc-binding site, a lipophilic tube connecting the surface with the zinc-binding site, as well as a catalytic zinc ion. HDAC-2 may be post-translationally modified by phosphorylation, acetylation, ubiquitination, and sumoylation, while it exhibits high activity and enantioselectivity to histones [8]. By catalyzing the removal of acetyl groups on the NH2-terminal lysine histone residues, HDAC-2 is involved in transcriptional repression and tumor-suppressor gene-silencing [2,9]. As a consequence, its deregulation may potentially promote malignant cell proliferation, migration, and/or invasion [10].

HDAC2 plays a vital role in gene expression through the formation of transcription repressor complexes. It is often regarded as a target for cancer therapy [5]. Increased deacetylation can cease transcription of tumor-suppressor genes leading to cell proliferation, migration, and invasion [11]. Different subtypes of HDAC seem to be related to different cancer biological behaviors and histological types. It is a component of complexes that deactivate the SIN3 and NURD pathways [12]. In addition, it plays a preventive role against neural expression genes in tissues unrelated to the nervous system [13]. Finally, HDAC-2 is regarded as a helpful factor in vascular and lymphatic invasion in a variety of malignancies [5]. Another study showed, through immunohistochemistry, that HDAC-2 expression was related to the presence of both lymphatic and vascular invasion and lymph node metastases in malignant thyroid lesions. Additionally, both nuclear and cytoplasmic pattern of HDAC-2 distribution is related more to lymphatic invasion [14].

Histone deacetylase inhibitors (HDACI) are enzymes that lead to hyperacetylation of the substrates. Four classes of HDACI, which differ in structure, are being researched. These include hydroxamic acid (suberoylanilide hydroxamic acid, SAHA), cyclic tetrapeptide (romidepsin), benzamide (MS-275), and aliphatic acid (valproic acid) [5,15]. Another classification, which can be also used, depends on the specificity of each HDACI to the HDAC. For example, MS-275 and romidepsin inhibit class I, while trichostatin A and SAHA are pan-HDAC inhibitors [5,16]. They suppress the action of HDAC and express anti-tumor activity through many mechanisms. These include both the damage to and prevention of repairment of DNA via the production of reactive oxygen species (ROS) [17]. The HDACIs can alter gene expression and negatively affect cell proliferation in various stages of the cell cycle, such as the G1, G2, or M phase [18]. Additionally, apoptosis can be triggered through both the endogenous and exogenous pathways by the HDACIs [19]. The HDACIs exert an antiangiogenetic and antimetastatic effect on malignancies, as well as prevent malignant cells from using glycose for their metabolic pathways [20,21,22]. It is of paramount importance to highlight that their action is limited to the malignant cells, whereas normal ones are only slightly affected. Clinical research on the use of HDACIs has been conducted and proved their antitumor potential [5,15,23,24]. Furthermore, four HDACI have already been approved by the FDA (U.S. Food and Drug Administration) against haematological malignancies: T-cell cutaneous lymphoma is being treated with romidepsin (istodax) or vorinostat (zolinza); T-cell peripheral lymphoma is being treated with belinostat (beleodaq); while panobistat (farydak) is being used against multiple myeloma [5]. In breast cancer (BC), HDACIs show clinical benefits when used as a combination therapy with radiation or cytotoxic treatment [25].

BC is the most frequent malignancy affecting the female population worldwide [26,27,28,29]. As much as 30% of all cancers derive from the breast tissue and it is the second cause of cancer death in women following lung cancer in developed countries [30,31]. Diverse therapeutic methods are currently being used to treat patients, including surgery, radiotherapy, and chemotherapy [32,33,34,35]. These methods do have side effects, including toxicity and drug resistance [36]. Additionally, despite the extensive advancements in both treatment and diagnosis, 43,250 women are, however, still estimated to die in the US in 2022 [31]. Hence, it is necessary to explore novel aspects regarding the treatment and prognosis of this specific disease.

HDAC-2 has been reported to play a significant role in various cancer entities, ranging from medulloblastoma, melanoma, lung cancer, and hematological malignancies, to pancreatic, colorectal, prostate, and urothelial cancer [37]. Recently, multiple study groups have investigated the role of HDAC-2 in BC and underlined its potential oncogenic capacities in different BC types [38,39,40,41,42]. Nevertheless, most of these studies investigate HDAC-2 expression in BC cell lines or xenograft tumor models, but not patient-derived tumor samples. Thus, the aim of our study is to examine the expression of HDAC-2 immunohistochemically in specimens derived from BC tissue and its correlation to the clinicopathological features of the tumor and patient prognosis. 

During the last three decades, great progress on the pathophysiology of BC has been achieved through extensive research, leading to the development of possible novel therapeutic approaches. In addition, a variety of human malignancies, such as lung, colorectal, gastric, thyroid, prostate, endometrial, hepatocellular, kidney, pancreatic, melanoma (including uveal), breast, and hematological malignancies, over-express HDACs [15,25,33,34,35,36,37,38,39,40,41,42,43,44,45,46,47,48,49,50,51,52,53,54,55]. The majority of these studies have highlighted the correlation between HDAC over-expression and parameters such as the histological grade, the stage of the tumor, and the survival of the patients. Still, the available knowledge about HDACs and BC remains poor.

## 2. Materials and Methods

### 2.1. Clinical Material

Patient-derived BC tumor samples with a diameter less than 20 mm were obtained from female patients that had undergone R0 surgical resection (R0 resection: no cancer cells seen microscopically at the margin of the resection) of breast tissue during the period 2008–2018. Only deceased, pre-operatively treatment-naïve, stage I–III BC patients were included in the study. Stage IV BC patients, patients with tumor size >20 mm, neoadjuvantly-treated patients, and patients whose death was associated with any reason not directly linked to BC were excluded from the study. The staging of the tumor was evaluated using the 7th edition of the American Joint Committee on Cancer (AJCC) Grouping system and the Tumor, Node, Metastasis (TNM) system [56]. A total of 118 patients meeting the aforementioned criteria were selected to be evaluated in our study (Figure 1). The period of time from the date of surgery until death caused by BC was defined as the overall survival (OS) period. Disease-free survival (DFS) period was defined as the time interval between initial diagnosis and disease recurrence. All patients included in this study had given written informed consent for the evaluation of their clinical information and biological specimens. The study was approved by the Ethical Committee of the Medical School of the National and Kapodistrian University of Athens (Approval ethic code: 1718004914). 

### 2.2. Immunohistochemistry

BC tissue samples were initially paraffin-embedded, and formalin-fixed. Rabbit polyclonal anti-HDAC-2 (H-5, Santa Cruz Biotechnology, Santa Cruz, CA, USA, sc-7899) antibodies were used in order to examine HDAC-2 expression immunohistochemically. The antigen was retrieved through the use of microwave slides for 15 min in 10 mM citrate buffer, in compliance with the manufacturer’s instructions. Hydrogen peroxide 0.3% was mixed with methanol for 30 min, in the dark at room temperature, in order to achieve removal of the endogenous peroxidase activity. Then, the incubation of all sections, at room temperature for 1 h with antibodies against HDAC-2 (H-54, sc-7899, Santa Cruz Biotechnology) in phosphate-buffered saline (PBS - Primary Antibody Diluent, ScyTek Laboratories Inc., Logan, UT, USA) dilution 1:200, took place. Two more incubations at room temperature lasting 10 min each, one with a biotinylated linking reagent and peroxidase-conjugated streptavidin label, followed. A 3,3′-diaminobenzidine tetrahydrochloride (DAB) substrate kit (UltraVision Quanto HRP Detection System, Thermo Fisher Scientific, Labvision Corporation, Fremont, CA, USA) was then used in order to develop immune peroxidase activity. The sections were stained with hematoxylin. Negative controls for the study were performed through the use of irrelevant anti-serum or omission of the primary antibodies, whereas positive controls were tissue parts from pancreatic cancer with previously known increased levels of HDAC-2 [57]. The cells’ proliferative index of the tumor was evaluated by p53 immunohistochemical expression.

### 2.3. Evaluation of Immunohistochemistry

Two pathologists, unaware of the clinical information, assessed the sections immunohistochemically by measuring at least 1000 malignant cells each time. Two parameters were used in order to evaluate the immunohistochemistry staining. These were the intensity (0: negative, 1: mild, 2: moderate, 3: strong) and the percentage of positive cells (0: negative staining; 1: less than 10%; 2: equal to or more than 10% and less than 33%; 3: equal to or more than 33% and less than 66%; 4: equal to or more than 66%). The HDAC-2 immunohistochemistry scores were then measured through the multiplication of these two parameters. The examined cases were divided into two groups depending on the above score: 0–6 was characterized as low expression of HDAC-2, and 7–12 as high expression of HDAC-2. The staining of p53 was regarded as positive when the percentage of positively stained tumor nuclei was more than 10%.

### 2.4. Statistical Analysis

Quantitative variables were expressed as mean values (SD) or as median values (interquartile range = IQR). Qualitative variables are presented with absolute and relative frequencies. For the comparisons of proportions, chi-square and Fisher’s exact tests were used. Student’s t-tests were computed for the comparison of mean values in the case of normal distribution, and the Mann–Whitney test for the comparison of median values when the distribution was not normal. Kaplan–Meier survival estimates for events were graphed over the follow-up period. Log-rank tests were used to compare survival curves. The Cox proportional hazard model was used in order to determine independent factors for recurrence and survival [58]. Hazard ratios (HR) with 95% confidence intervals (95% CI) were computed from the results of the Cox regression analyses. Statistical significance was set at 0.05 and analyses were conducted using the SPSS statistical software (SPSS 22.0; SPSS Corporation, Chicago, IL, USA).

## 3. Results

Data from 118 women were analyzed. Sample demographics and clinical characteristics are presented in Table 1. The mean age of the women was 63.7 years (SD = 8.2). A total of 32.2% of the cases were grade 3 and 16.9% of the samples were triple negative. Moreover, 33.9% of the cases were at stage II and 49.2% of the cases were at stage III. A total of 72.9% were estrogen receptor (ER)-positive and 67.8% were progesterone receptor (PR)-positive. C-erb B-2 was positive in 15.3% of the women. In terms of postoperative adjuvant treatment, all patients received radiation therapy, as well as adjuvant hormone therapy, in case of hormone receptor-positive BC. Depending on the individual case, patients with HER2-positive BC additionally received trastuzumab-based targeted therapy, as well as adjuvant chemotherapy. Patients with tumors larger than 1 cm in diameter, and/or hormone receptor-negative BC, also received 5-fluorouracil PLUS epirubicin PLUS cyclophosphamide, followed by docetaxel. 

Only 4 (3.4%) out of 118 cases did not show HDAC-2 expression (Figure 2A). In all other HDAC-2-positive cases, the pattern of distribution was nuclear (Figure 2B,C), whereas non-cancerous sites of the tissues were negative for HDAC-2. The mean immunohistochemical expression of HDAC-2 was 5.02 (SD = 2.71), with 22 cases (18.6%) exhibiting high immunohistochemical expression (Figure 2C).

The association of high immunohistochemical expression with demographics and clinical characteristics is shown in Table 2. High HDAC-2 expression correlated with the lobular type and stage/grade III BC. Moreover, the proportion of cases with high HDAC-2 expression was higher in ER- and C-erb B-2-positive samples.

The mean DFS for all patients was 28.6 months (SD = 1.8), with a median DFS of 21 months, while the mean OS was 42.2 months (SD = 2.0), with a median OS of 40 months. Kaplan–Meier estimates for both DFS and OS were significantly different for low versus high HDAC-2-expressing BC (log-rank test, *p* < 0.001) (Figure 3).

Multiple Cox regression analysis for DFS (Table 3) identified high immunohistochemical expression as an independent prognostic factor, associated with greater hazard after adjustment for age, histological type, grade, stage, ER, PR, and C-erb B-2. Specifically, patients with high HDAC-2 expression had 3.31 times greater hazard for progression than those with low expression (*p* < 0.001).

Multiple Cox regression analysis for survival (Table 4) showed a 2.56 times greater hazard for those with high immunohistochemical expression (*p* < 0.001) after adjusting for age, histological type, grade, stage, ER, PR, and C-erb B-2.

## 4. Discussion

As aforementioned, HDACs are currently regarded as crucial factors in the regulation of the cell cycle, including cell differentiation, apoptosis, and proliferation, in different types of cancer [15,25,41,43,44,45,46,47,48,49,50,51,52,53,54,55]. Their over-expression correlates with the management and survival of the patient.

In our study, we examined the clinical importance of the expression of HDAC-2 in 118 human BC cases. Our results indicated that high levels of HDAC-2 were associated with the lobular histological type of BC (*p* < 0.005). ER- and C-erbB-2-positive specimens were more likely to present higher HDAC-2 levels than PR-positive or triple-negative specimens. Moreover, there was a strong correlation between stage III (*p* < 0.001) and histological grade 3 (*p =* 0.013), and HDAC-2 over-expression. This indicates a more aggressive tumor behavior with a worse prognosis. Last but not least, DFS and OS were negatively associated with the over-expression of HDAC-2, thus confirming the above findings about prognosis. 

A small number of studies have so far evaluated the clinical importance of HDAC-2 in BC. Derr et al. indicated that only the combination of high SIRT-1, LSD1, and HDAC-2 leads to shorter survival and a greater possibility of relapse. The over-expression of HDAC-2 alone did not show any clinical significance [59]. The same year, Seo et al. demonstrated that high levels of HDAC-2 correlate with better OS in ER-negative tumors [60]. Different environments or sizes of specimens, as well as different ethnicities of patients (Caucasian or Asian), could explain the discordance between our findings and these results. The number of patients also undoubtedly represents an important factor explaining the difference in the results. The study by Derr et al. included 822 [59], and the one by Seo et al., 300 patients [60], whereas our study evaluated the BC tumor samples of a total of 118 patients. In addition, different antibodies, methods of staining, including dilution levels, and even the evaluation of the immunohistochemistry could justify the diversity of the results. In our study, evaluation was performed through the use of both the intensity and the percentage of the stained cells. It should be highlighted that every study had different inclusion and exclusion criteria concerning age, stage, histological type, and the number of lesions. This could result in heterogeneity of the population among the studies and explain the aforementioned conflicting results. On the other hand, several studies have demonstrated similar results concerning HDAC-2 expression and its association with BC. In 2013, Muller et al. showed that HDAC-2 is associated with more aggressive BC and lymph node metastasis [61]. In 2016, a Chinese study concluded that higher levels of HDAC-2 are related to worse clinical outcomes in BC patients, as metastases, disease recurrence, and drug resistance occur earlier. Specifically, HDAC-2 overexpression was found to correlate with anthracycline resistance, lymph node metastasis, Ki-67 expression, advanced TNM stage, and higher histological grade in BC [62]. One year later, another study demonstrated the same results, since the over-expression of HDAC-2 was positively associated with lymph node infiltration, undifferentiated cancer cells, and a worse clinical outcome [63]. Altogether, the present study confirmed the results of previous research groups and determined important clinicopathological parameters that correlate with the over-expression of HDAC-2 in BC, such as higher histological grade, stage, and worse prognosis.

In the same context, targeting of HDACs by the use of HDAC inhibitors (HDACIs) might represent a novel therapeutic strategy against BC [25,54,64]. In vitro studies have shown that HDACIs do improve response to hormone therapy and increase estrogen receptor expression in cancer cells [65]. In a clinical trial, Munster et al. demonstrated that the combination of tamoxifen with the HDACI vorinostat has better clinical results than tamoxifen alone. Tamoxifen normally exerts its anti-tumor action through cessation of proliferation. However, its combination with HDACI increases its apoptotic activity [66]. Specifically, this action is accelerated through the inhibition of HDAC-2 [67]. Another study proved that a natural HDACIs can reduce HDAC-2 levels in BC cell lines and lead to apoptosis and inhibition of tumor cell proliferation [68]. Finally, Xu et al. recently proposed HDAC-2 as a target of anti-cancer immunotherapy in triple-negative BC [38]. All in all, the HDAC/HDACI interaction is a complex issue that needs to be clarified in further studies.

## 5. Conclusions

In summary, this study examined the expression of HDAC-2 in 118 deceased sporadic BC patients and its correlation with clinicopathological characteristics of the tumor and the prognosis of the patient. High expression of this protein is associated with higher histological grade, stage of disease, and worse prognosis. Although more research needs to be conducted, our study indicates that HDAC-2 could arise as a new potential index of aggressiveness and a therapeutic target against BC. Except for the HDAC-2, future studies need to closely investigate the mechanisms of action of different members of the HDAC family in BC, and to determine the tissues mostly affected. This is a prerequisite for the future development of effective HDACIs for the efficient treatment of the various BC subtypes.

## Data Availability

The data presented in this study are available on request from the corresponding author.

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
