# Peer review of "Clinical Significance of the Histone Deacetylase 2 (HDAC-2) Expression in Human Breast Cancer"

_jpm, 2022, doi:10.3390/jpm12101672_

Round 1

Reviewer 1 Report

In this investigation Garmpis et al. measured the clinical significance of histone de-acetylase 2 (HDAC-2) in breast cancer (BC). They explored the relation between HDAC-2 and clinico-pathological features in 118  specimens obtained from breast cancer surviving patients diagnosed with  tumor size <20 mm corresponding to the stage I, II and III. They demonstrated Higher HDAC-2 expression was related to lobular histological 31 type of cancer, grade III and stage III along with shorter rates of disease-free and overall survival attributes. However, Authors need to address the following concerns.

1)    The introduction is not well drafted, specific emphasis on HDAC-2 is recommended to gain readers attention more.

2)    What’s the fate of other HDAC family members in these 118 specimens of the patients. State the rationale behind the choosing of HDAC-2 among other members in this study.

3)    Did you evaluated the expression patterns of HDAC-2 in ductal carcinoma models?

4)    The DFS and OD results in Figure 3, reflects the data from patients with lobular cancer? If so, what are the DFS and OS rates in patients with ductal cancers.

5)    The quality of figure 3 is poor and can be improved and text inside the figure should be manually written. Also, the image labeling is not included and can be rectified.

6)    Can HDAC-2 as used as prognostic marker for diagnosing the patients with lobular breast cancer? Also, can it be chosen as biomarker to detect the disease?

7)    The manuscript can be further revised for grammatical and typological errors.

_ Minor

Inconsistencies over table numbering. Represented in Roman in text while in table  it was in numerical forms.

Reviewer 2 Report

The authors investigated the clinical importance of histone deacetylases (HDAC)-2 in breast cancer. They concluded that higher HDAC-2 expression was related to lobular histological type of cancer, grade III and stage III. Moreover, the disease-free and overall survival were shorter and negatively related to the over-expression of HDAC-2, respectively. However, the authors cannot give general inferences because the sample size is relatively small. The subject and purpose of the study was clinically important, but there are some points that need to be discussed and clarified.

 Reviewer’s comments

1.           All of the cases had tumors that were smaller than 2 cm and almost half of the cases were stage III. The authors should show the lymph node status. Furthermore, the relationship between the lymph node status and HDAC-2 overexpression should also be shown.

2.           The authors stated that ER and C-erbB-2 positive specimens were more likely to present higher levels of HDAC-2 compared to the PR positive or triple negative specimens. PR positive tumors are a different category from TN tumors. Moreover, the authors described that there was a strong correlation between stage III (p<0.001) and histological grade 3 and HDAC-2 overexpression. In this study, the authors did not indicate whether ER positive cases were at risk for recurrence.

3.           The authors stated in the Introduction section that HDAC-2 is regarded as a helpful factor of vascular and lymphatic invasion (LVI) in a variety of malignancies. They should show the relationship between LVI and HDAC-2 expression.

4.           The authors should include information on postoperative adjuvant therapy.

5.           The authors should consider the relationship between HDAC-2 overexpression and hormone sensitivity.

Round 2

Reviewer 1 Report

Satisfied with Author's response.

Reviewer 2 Report

The author duly responds to my comment.